# Effect of Mycophenolate Mofetil Therapy on Recurrence of Hepatocellular Carcinoma after Liver Transplantation: A Population-Based Cohort Study

**DOI:** 10.3390/jcm10081558

**Published:** 2021-04-07

**Authors:** Yung-Fong Tsai, Fu-Chao Liu, Chun-Yu Chen, Jr-Rung Lin, Huang-Ping Yu

**Affiliations:** 1Department of Anesthesiology, Chang Gung Memorial Hospital, Taoyuan 333, Taiwan; l12084@cgmh.org.tw (Y.-F.T.); ana5189@cgmh.org.tw (F.-C.L.); an5376@cgmh.org.tw (C.-Y.C.); 2College of Medicine, Chang Gung University, Taoyuan 333, Taiwan; 3Clinical Informatics and Medical Statistics Research Center, Chang Gung University, Taoyuan 333, Taiwan; jr@mail.cgu.edu.tw; 4Graduate Institute of Clinical Medicine, Chang Gung University, Taoyuan 333, Taiwan

**Keywords:** hepatocellular carcinoma, immunosuppressant, liver transplantation, recommended defined daily dose, recurrence, mycophenolate mofetil, population-based study

## Abstract

Hepatocellular carcinoma (HCC) recurrence after liver transplantation is associated with immunosuppressants. However, the appropriate immunosuppressant for HCC recipients is still debated. Data for this nationwide population-based cohort study were extracted from the National Health Insurance Research Database of Taiwan. A total of 1250 liver transplant recipients (LTRs) with HCC were included. We analyzed the risk factors for post-transplant HCC recurrences. Cumulative defined daily dose (cDDD) represented the exposure duration and was calculated as the amount of dispensed defined daily dose (DDD) of mycophenolate mofetil (MMF). The dosage effects of MMF on HCC recurrence and liver graft complication rates were investigated. A total of 155 LTRs, having experienced post-transplant HCC recurrence, exhibited low survival probability at 1-, 3-, 5-, and 10-year observations. Our results demonstrated increased HCC recurrence rate after liver transplantation (*p* = 0.0316) following MMF administration; however, no significant increase was demonstrated following cyclosporine, tacrolimus, or sirolimus administration. Notably, our data demonstrated significantly increased HCC recurrence rate following MMF administration with cDDD > 0.4893 compared with cDDD ≤ 0.4893 or no administration of MMF (*p* < 0.0001). MMF administration significantly increases the risk of HCC recurrence. Moreover, a MMF-minimizing strategy (cDDD ≤ 0.4893) is recommended for recurrence-free survival.

## 1. Introduction

Hepatocellular carcinoma (HCC), which causes 781,637 mortalities annually, is the third leading cause of cancer-related deaths worldwide [1,2,3]. Surgical removal may cure well-localized HCC and provide a higher long-term survival probability than conservative treatments. For early-stage unresectable HCC or that complicated with end-stage liver disease, liver transplantation is no longer considered a contraindication and has been recognized as a definitive treatment option for complete removal of the carcinogenic liver [4,5]. In Taiwan, the first transplantation for treating a patient with HCC was performed in 1999. Afterwards, transplant surgery for HCC treatment gradually increased. Liver transplantation for HCC eradication presents a medical challenge of oncological relapse, which is negatively affected by long-term immunosuppression. Unfortunately, HCC recurrence within 2 years of transplantation has been noted in 15–20% of liver transplant recipients (LTRs) [6], and it remains the major cause of post-transplant death in LTRs with HCC.

Selection criteria for transplant candidates with favorable HCC should be carefully identified [7,8]. The Milan criteria have been the generally recommended standard since 1996 [7], and 5-year survival achieved using this criterion is higher than that obtained using other criteria. Several selection indicators, including tumor invasion, differentiation, number, and size, have been reported as potential markers [9,10,11]. Daily immunosuppressive agents are indispensable for reducing immunological injury to immunologically nonidentical allograft and for preventing graft rejection and functional failure. However, tumorigenesis progresses more aggressively and rapidly under immunosuppressed state after transplantation. Immunosuppressants upregulate the HCC recurrence rate and reduce the long-term survival probability [12]. There are few available clinical studies on immunosuppressant dosage and selection for reducing HCC recurrence risk.

Immunosuppressive regimens, including steroids, calcineurin inhibitors, antimetabolite purine antagonists, and mTOR inhibitors, contain various combinations of drug classes that differ in their mechanism of action and that effectively reduce severe drug toxicity [13,14]. Calcineurin inhibitor is the first-line immunosuppressant for liver transplantation; however, its contribution to HCC recurrence with a dose-dependent relationship has been demonstrated [15,16,17]. Calcineurin inhibitors have also been reported to cause nephrotoxicity and neurotoxicity among transplant recipients [18,19]. Mycophenolate mofetil (MMF), an antimetabolite purine antagonist, has been introduced as an adjunctive combination regimen in LTRs with renal failure or encephalopathy and as an agent to minimize nephrotoxicity and neurotoxicity caused by calcineurin inhibitors. The mechanism of MMF, causing inhibition of clonal expansion of lymphocytes activated by transcription factors, is distinct from that of calcineurin inhibitors [20]. Immunosuppressant therapy should be able to balance the protective effects for both graft survival and HCC recurrence risk.

Several studies on recurrent HCC in LTRs after transplantation have been conducted, most of which have small cohort populations or single-center design. This nationwide cohort research retrospectively investigated the risk factors for post-transplant HCC recurrence over 14.5 years. Immunosuppressive therapy for LTRs has been modified over the past decade. The approach for minimizing steroid and calcineurin inhibitor exposure through the addition of MMF to the regimen should be reassessed for HCC recurrence risk. The aim of our nationwide cohort study was to retrospectively analyze the effect of MMF therapy on HCC recurrence and other clinical variables after liver transplantation.

## 2. Materials and Methods

### 2.1. Data Source and Study Participants

The study population was identified from Taiwan’s National Health Insurance Research database (NHIRD) between 1998 and 2012. Nationwide, 23.3 million residents were enrolled in the NHI mandatory program aimed to cover more than 99.9% of beneficiaries by the end of 2012. The NHI program was developed in 1996, with which almost all patients were registered, and the record is available in the NHIRD. All claim information from the NHIRD was encrypted and outputted in a computerized format for research purposes. Information obtained from the NHIRD included demographic data, prescription drugs, and procedures. The diagnostic codes of diseases were encoded using International Classification of Disease, Revision 9, Clinical Modification (ICD-9-CM). All diagnoses and LTRs enrollment in the NHI program were executed by a transplant surgeon or a gastroenterologist. Otherwise, the annual incidence rate of HCC in the general population was obtained from the Taiwan Cancer Registry database.

This cohort study was approved by the Institutional Review Board of Chang Gung Memorial Hospital (Registration number: IRB 103-0102B) and the NHIRD research committee (Registration number: NHIRD-103-103). Since information of all individuals was anonymized and de-identified, the requirement of obtaining informed consent was waived.

### 2.2. Identification of LTRs with Preoperative HCC

LTRs with pretransplant HCC were included in this cohort study. LTRs were recognized using the ICD-9-CM code 996.82 or V427 from the catastrophic illness database between July 1998 and December 2012. From the targeted LTRs, only those patients who had received liver transplant surgery (code 505, 75020A, or 75020B) within the study period were included (Figure 1). Furthermore, patients identified as target candidates exhibited HCC before liver transplantation. The configured ICD-9-CM codes for identification of preoperative HCC were 155.0, 155.2, and 197.7. Moreover, LTRs claimed to have developed cancer within 3 months of liver transplantation were allocated to preoperative malignancy group due to inadequate malignancy survey or insensitive preoperative studies before transplantation.

Disease diagnosis was established if the claimed codes for diagnosis were recorded at least 5 times in the outpatient department or once in the inpatient department. Death identification was established as death code claimed or the termination of NHI program.

### 2.3. Definition of HCC Recurrence

The present study evaluated the risk of post-transplant HCC recurrence in the enrolled population. Primary outcomes were pretransplant HCC recurrence and long-term mortality in the LTRs. Diagnosis of post-transplant HCC recurrence was established according to ICD-9-CM codes 155.0, 155.2, and 197.7 in the database after transplant surgery. HCC recurrence was defined as the relapse of HCC after surgical removal of the recipient’s liver. Patients yielded a malignancy remission interval supported by negative HCC detection. Other related variables used to compare the recurrent risk between study groups were demographic data and comorbidities. Cyclosporine, tacrolimus, MMF, and sirolimus were other major immunosuppressants of interest used in the analysis.

Long-term survival outcomes of LTRs with or without recurrence were calculated as the period between transplantation and death. One of the groups, in which preoperative HCC did not revert after transplantation, represented the HCC-cured group. By contrast, another group, in which HCC relapsed postoperatively, represented the HCC-recurrent group.

### 2.4. Dosage of MMF Exposure

To observe HCC recurrence, post-transplant prescriptions for immunosuppressants in the outpatient and inpatient care order files during the index date were identified. Data such as MMF dosage per prescription, number of days supplied, and daily dosage were collected from the database. To measure the prescribed MMF dosage, we used the WHO-recommended defined daily dose (DDD) of 2 g (via oral or parenteral route) as the standardized unit [21]. DDD represents the recommended average dosage for daily maintenance to treat a disease in adult patients. We compared the MMF dosage effects on the basis of the same standard by calculating the numbers of DDDs. The number of DDDs was calculated as the total amount of doses divided by the amount of MMF in a DDD. Cumulative DDD represented the exposed duration and was calculated as the amount of dispensed DDD of MMF. We determined the effect of MMF on the HCC recurrence risk by estimating the average drug concentration during the observation period. The observation period was defined as the period between transplantation and HCC recurrence or death in recurrent LTRs or until the end of 2012 in nonrecurrent LTRs. The average MMF concentration was weighted as cumulative DDDs divided by the length of the observation period. To analyze the dose–effect relationship, LTRs in the study cohort were allocated into 3 groups on the basis of average MMF concentration, namely unused MMF, low MMF dosage (≤0.4893 DDD/per day), and high MMF dosage (>0.4893 DDD/per day). The cutoff value of the safety threshold determined using the Cox model analysis method was 0.4893 DDD/per day.

### 2.5. Statistical Analysis

We used the chi-square test or Fisher exact test for analyzing categorical variables and the Student *t* test for analyzing numerical variables to compare risk factors between groups. The Kaplan–Meier method (SAS Institute Inc.; Cary, NC, USA) was used to compute HCC recurrence cumulative incidences and long-term mortality. The Cox model was used to examine the hazard ratio after adjustment of variables. All statistical analyses were performed using SAS statistical software (version 9.3; SAS Institute Inc.; Cary, NC, USA). A 2-tailed *p*-value of <0.05 was considered significant.

## 3. Results

### 3.1. Characteristics of the Study Cohort

A total of 2938 LTRs who underwent liver transplantation were included in this study, of whom 2068 were men and 870 were women. After careful screening for malignancy, 1415 LTRs were encoded as having preoperational malignancy, and transplantation was indicated in 1250 (88.34%) of them with HCC. The remaining 1523 LTRs were encoded as having no preoperative malignancy. Of the 2938 LTRs who underwent transplantation during 1998–2012, 1250 (52.13%) patients were diagnosed as having end-stage liver disease with preoperative HCC (Figure 1). Among them, 970 (77.6%) were men and 280 (22.4%) were women; 151 (12.08%) exhibited HCC recurrence after liver transplantation, and 1099 (87.92%) were cured of HCC until the end of 2012. The average age at the time of transplant surgery in this cohort was 53.56 years, which was higher than the average age of the study cohort (*n* = 2398) consisting of LTRs other than those with HCC (46.42 ± 17.68 years) [22]. The average observation period of this cohort was 3.79 ± 3.26 years, with 14,176 person-years of follow-up in total. In this cohort, liver cirrhosis was observed in 1145 patients. Among 1240 LTRs with chronic hepatitis, HBV-, HCV-, and alcohol-related incidences were observed in 755, 377, and 211 patients, respectively. Additionally, comorbidities including renal failure (*n* = 41), cardiovascular disease (*n* = 49), cardiac artery disease (*n* = 110), pulmonary disease (*n* = 214), hypertension (*n* = 310), and diabetes mellitus (*n* = 324) were observed in the LTRs (Table 1).

### 3.2. Survival Probability Was Reduced in LTRs with Post-Transplant HCC Recurrence

The incidence of de novo liver malignancy in the Taiwanese population is increasing gradually per year; it increased from 0.0325% in 1998 to 0.0490% in 2012 (Figure 2). In the cohort of 1250 LTRs with HCC, 155 (12.4%) recipients exhibited HCC recurrence during the index period (Table 2). The average time of diagnosis of recurrence was 635.66 ± 482.93 days. Survival probability between HCC-recurrence and HCC-cured groups was compared using the Kaplan–Meier analysis method within the 14.5-year observation period (Figure 3). Our data indicated that the survival probability of LTRs with postoperative HCC recurrence was lower than that of those with cured HCC (*p* < 0.0989).

Overall mortality rates were 7.04% (88/1250), 11.68% (146/1250), 13.01% (163/1250), and 14.16% (177/1250) at 1-, 3-, 5-, and 10-year observations, respectively. The cumulative mortality rates in patients without HCC recurrence were 7.15%, 10.86%, 11.58%, and 12.56% at 1-, 3-, 5-, and 10-year observation, respectively, whereas in patients with HCC recurrence, the rates were 6.25%, 17.65%, 23.45%, and 25.83%, respectively. One-year patient mortality was similar between the groups; however, the 3-, 5- and 10-year survival probabilities were significantly inferior in the HCC-recurrent group. These results indicate a strong association between survival and HCC recurrence after transplantation, with higher mortality in patients with recurrence than in those without recurrence.

### 3.3. Risk Factors for Post-Transplant HCC Recurrence among the LTRs

To evaluate the contribution of probable risk factors, the LTRs with pretransplant HCC were allocated into two groups on the basis of their post-transplant recurrence experience within the cohort interval. Finally, post-transplant HCC recurrence and nonrecurrence groups consisted of 151 and 1099 LTRs, respectively. Table 1 shows the risk factors for HCC recurrence in both groups. Results revealed that age and sex were not significantly associated with the post-transplant HCC recurrence risks (*p* = 0.9516 and 0.9455, respectively). Moreover, medical conditions, such as liver cirrhosis, hepatitis B, and hepatitis C were not likely to contribute to HCC recurrences (*p* = 0.7696, 0.7442, and 0.5113, respectively) in the recipients. Other systemic diseases, such as hypertension, mellitus diabetes, lung disease, cardiovascular disease, or renal failure, also did not contribute to post-transplant HCC recurrence.

We further investigated the effects of four commonly used immunosuppressants, namely cyclosporine, tacrolimus, MMF, and sirolimus, on the cohort population. Among 1250 patients with HCC who received transplantation, 12.24%, 96.08%, 81.28%, and 26.72% had used cyclosporine, tacrolimus, MMF, and sirolimus, respectively. Interestingly, only MMF was found to significantly increase the HCC recurrence rate in our cohort (*p* = 0.0316). We further compared the postoperative HCC recurrence time in LTRs. The recurrence times in MMF-treated and non-MMF-treated LTRs were 640.40 ± 496.32 and 595.69 ± 360.24 days, respectively; however, the difference was not statistically significant (*p* = 0.7275).

### 3.4. MMF Concentration Correlates to HCC Recurrence in the LTRs after Transplantation

Among the 1250 LTRs with HCC, MMF administration for post-transplant immunosuppression was documented in 1016 recipients and not in the remaining 234 recipients. We compared many probable factors contributing to HCC recurrence between groups with or without documented MMF administration. Physicians’ preference to prescribe MMF was found to be significantly higher in patients with liver cirrhosis (*p* < 0.0001); however, no significant difference in the preference to prescribe MMF was reported in patients with hepatitis B-, hepatitis C-, or alcohol-related chronic hepatitis (Table 3).

In this cohort, 1016 LTRs were administered MMF for post-transplantation immunosuppression. The mean dosage of MMF was 355.15 ± 344.86 DDDs. Among them, 135 LTRs exhibited HCC recurrence, and 881 exhibited no recurrence after transplantation. The mean dosages of MMF in HCC recurrence and nonrecurrence groups were 390.67 ± 303.81 and 349.71 ± 350.56, respectively (*p* = 0.1550).

To evaluate the effect of MMF concentration, we standardized the average MMF concentration during the observation period by calculating the average cumulative DDDs for the observation period. The minimal and maximal concentrations of MMF were 0.0001 and 0.9197 DDD/per day, respectively. The 25%, 50%, and 75% quantiles were 0.1874, 0.2911, and 0.4417 DDD/per day, respectively. The Cox model analysis method was used to define the safety threshold for MMF concentration, and the cutoff value was defined as 0.4893 DDD/per day. Although MMF prescription was preferred in LTRs with liver cirrhosis (*p* < 0.0001) (Table 3), no difference in the recurrence time between HCC recurrence and nonrecurrence groups was observed (hazard ratio (HR): 0.912, confidence interval (CI): 0.490–1.694) (Table 1). HCC recurrence rates in unused MMF, low-dose MMF (average MMF concentration ≤0.4893 DDD/per day), and high-dose MMF (average MMF concentration >0.4893 DDD/per day) groups were 7.34%, 13.86%, and 22.76%, respectively. We analyzed the recurrence rate in three groups exposed to variable MMF dose by using the Cox model analysis method after adjustment for liver cirrhosis. A significantly higher recurrence rate was observed in the high-dose group than in the other groups (*p* < 0.0001, HR: 2.234, CI: 1.503–3.319); however, the rate was not found to differ significantly between the low-dose and MMF nonuse groups (*p* = 0.2240; Figure 4 and Table 4).

### 3.5. Lower MMF Concentration Did Not Increase Graft Complication Rate Compared with the Higher Concentration Group

The cutoff value of average MMF concentration for HCC recurrence risk was set at 0.4893 DDD/per day in the cohort. We further evaluated whether the low MMF concentration group exhibited higher graft complications (ICD-9-CM code of 996.82) than the high MMF concentration group. The coded graft complications included rejection, infection, vascular disorders, and neoplasms. We compared the graft protective effects in three subgroups with different MMF dosages. Analyzed immunosuppressant combinations included no MMF, low-concentration MMF, and high-concentration MMF. In total, 435 graft-related complications were reported in 1250 LTRs during the observation period. Rejection rates in the no MMF, low-concentration MMF, and high-concentration MMF groups were 69 (29.49%), 300 (35.80%), and 66 (37.08%), respectively; however, the difference in rejection rates between the groups was not significant (*p* = 0.1582), which indicated that graft failure risk in LTRs is not influenced by an increase or decrease in MMF concentration.

## 4. Discussion

Annually, more than 11,000 HCCs are newly diagnosed in Taiwanese populations and the HCC incidence in 2012 was 0.049% (Table 2). The annual incidence rate of HCC is increasing gradually (Figure 2) and has become a major public health burden [1,23]. Liver transplantation is a definitive and well-established treatment option for patients with unresectable HCC. However, tumor recurrence remains a serious concern in transplantation medicine. We have previously reported that 3.34% of LTRs exhibit de novo malignancies, and the recurrence rate after transplantation in LTRs with pre-existing malignancies is 13.83% [22]. In addition, LTRs with post-transplant malignancies presented a significantly poor survival probability than LTRs without post-transplant malignancies. The present study reported a recurrence rate of 12.08% in LTRs with pretransplant HCC. We compared the mortality rate between all LTRs with and without post-transplant HCC recurrence, and lower survival probability was observed in LTRs with post-transplant HCC recurrence. Prognosis and the survival probability were found to be greatly affected in LTRs with HCC recurrence than in those without HCC recurrence at 1-, 3-, 5-, and 10-year observations (Figure 3). The high-risk population therefore requires a careful malignancy prevention program to reduce HCC recurrence and an early intervention to improve survival probability.

In our previous study, we reported 284 post-transplant malignancies, including 34.51% de novo and 65.49% recurrent malignancies, in Taiwanese LTRs during 1998–2012 [22]. The most common de novo malignancies, such as liver cancer (19.39%) and oropharyngeal cancer (19.39%), were infection-related; however, the most recurrent tumor was liver cancer (83.33%) in LTRs with pretransplant liver malignancy. Clinical variables such as old age, male sex, liver cirrhosis, and hepatitis B were the predisposing factors for post-transplant oncogenesis in LTRs. However, these risk factors were not found to correlate with HCC recurrence after transplantation. The risk factors for HCC recurrence in LTRs with pretransplant HCC seem different from those for other post-transplant malignancies. In addition, cardiovascular disease, liver cirrhosis, renal failure, pulmonary disease, hypertension, and diabetes mellitus were not found to contribute to HCC recurrence.

Several studies have reported the inhibition of immune surveillance by immunosuppressants, leading to upregulated HCC recurrence [24,25,26]. Immunosuppression therapy disrupts the integrity of immune defense against infection and oncogenesis control [27]. We further compared the effect of immunosuppressant therapy between the study case (with HCC recurrence) and control (without HCC recurrence). Our results indicated that MMF administration significantly promoted HCC recurrence after liver transplantation (*p* = 0.0316); however, cyclosporine (*p* = 0.0936), tacrolimus (*p* = 0.2458), and sirolimus (*p* = 0.1218) were not found to promote recurrence in the study cohort. Further analysis indicated no difference in recurrent time between the MMF-treated and untreated LTRs. This cohort study indicates an essential role of MMF in HCC recurrence in Taiwanese LTRs. Studies have reported that calcineurin inhibitors dose-dependently increase HCC recurrence [15,16,17]; however, our data contradict this finding. Cyclosporine administration was relatively low (12.24%) in the study cohort; less cyclosporine utilization and small cyclosporine using population size may account for the contradictory result. Additionally, tacrolimus utilization was common and high (96.08%) in both the groups, and tacrolimus therapy was received by almost every LTR; however, improper distribution of sample size in the two groups may account for the distinction.

Immunosuppressant-induced HCC recurrence increases in a dose-dependent manner [15,16,17]. Total MMF exposure in the present study was measured as the prescribed MMF dosage. The WHO-recommended DDD (2 g/per day) was considered a standardized unit. The total MMF dosage was higher in the HCC-recurrence group than in the nonrecurrence group (*p* = 0.1550). Subsequently, we standardized the average MMF concentration during the observation period by calculating the average cumulative DDDs during the period. We further defined the safety threshold for MMF concentration and defined the cutoff value at a concentration of 0.4893 DDD/per day. Physicians preferred to prescribe MMF to LTRs with liver cirrhosis (*p* < 0.0001) (Table 3). We analyzed the recurrence rate in three groups exposed to different doses by using the Cox model analysis method after adjustment for liver cirrhosis. Averaged daily dosage greater than 0.4893 DDD significantly increased HCC recurrence rate (22.76%) (*p* < 0.0001, HR: 2.234, CI: 1.503–3.319). However, minimization of the daily dosage to less than or equal to 0.4893 DDD in the low-dose MMF group resulted in a significantly lower HCC recurrence rate (13.86%) compared with that in the high-dose group. Moreover, the low MMF dose (≤0.4893 DDD) and MMF nonuse groups exhibited no significant difference in the recurrence rates. MMF concentration highly correlates to HCC recurrence in LTRs after transplantation. We further evaluated whether the episodes of graft adverse events would be increased by minimizing MMF concentration. Our data revealed no significant difference between groups (*p* = 0.1582). Minimizing MMF concentration had no effect on graft failure risk; however, it reduced HCC recurrence in LTRs in the study cohort. The most commonly used regimens include combinations of corticosteroid, calcineurin inhibitors, and antimetabolite purine antagonists [28]. Adjuvant MMF use is indicated for reducing the side effects related to steroid and calcineurin inhibitor. However, our results did not imply that high-dose mycophenolate mofetil had no protection of graft survival by immunosuppression. The veiled effects of MMF might be due to co-medication with corticosteroid, calcineurin inhibitors, or other immunosuppressants.

Although several retrospective and prospective studies have reported HCC recurrence risk following liver transplantation, most of them were limited by either a single-center design or short-term follow-up. The present nationwide cohort study was a large and long-term study, which allowed a more precise analysis of multivariate risk factors for post-transplant HCC recurrence. Our study presented some limitations. The NHIRD is a secondary database that does not contain several clinical data, including tumor morphological criteria, histo-pathological information, laboratory studies, physical examination, severity of comorbidities, and the relationship between disease and death. Additionally, evidence to support the correlation between MMF dose and plasma MMF level are not robust, and the relationship may be influenced by age, blood albumin level, and co-medications [29]. Therefore, interpatient variability in plasma MMF level may exist. Further prospective investigation is warranted to realize a therapeutic range of MMF levels in blood and to weigh the immune tolerance for immunosuppression-minimizing strategies.

## 5. Conclusions

Our study demonstrated lower survival probabilities in the group with post-transplant HCC recurrence than in the group with cured malignancy. Moreover, our data suggested that an MMF-minimizing strategy under a well-balanced combination of immunosuppressive agents is beneficial for patients with HCC who have undergone liver transplantation.

## Figures and Tables

**Figure 1 jcm-10-01558-f001:**
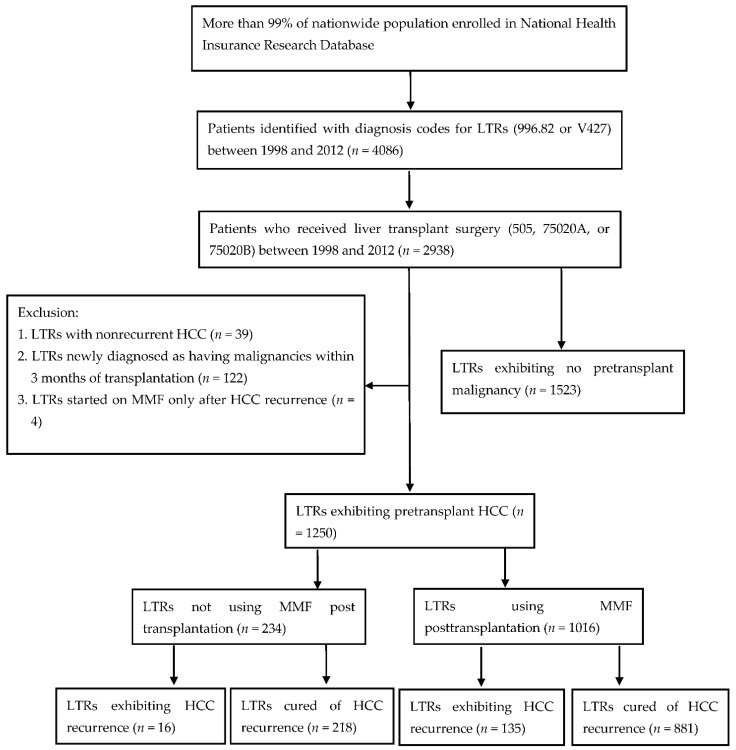
Flowchart of study selection. LTRs, liver transplant recipients; HCC, hepatocellular carcinoma; MMF, mycophenolate mofetil.

**Figure 2 jcm-10-01558-f002:**
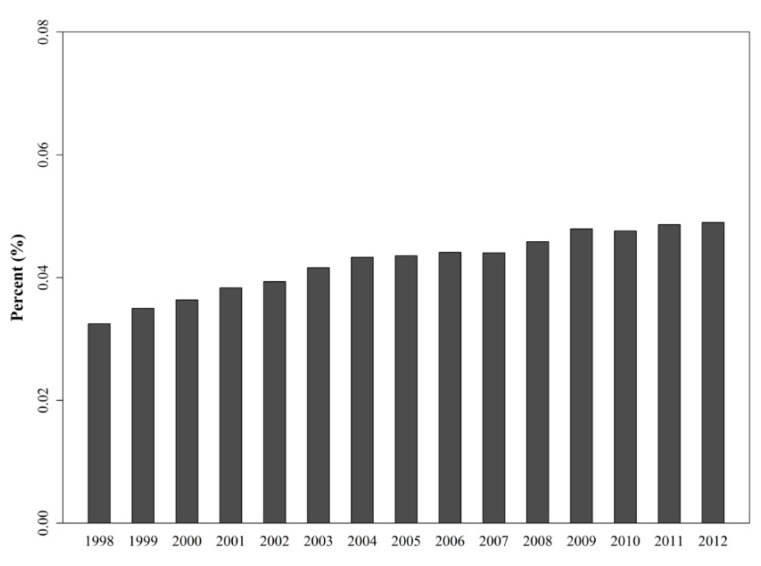
Annual incidence of de novo liver malignancy in general Taiwanese populations from 1998 through 2012.

**Figure 3 jcm-10-01558-f003:**
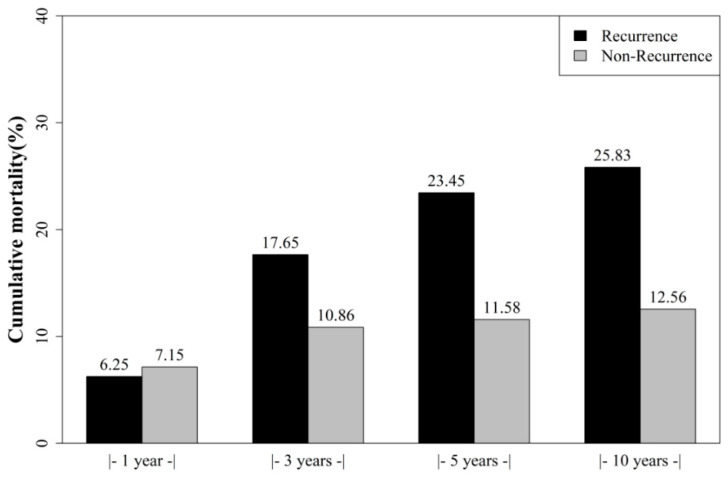
Cumulative probability of hepatocellular carcinoma recurrence in the liver transplant recipients.

**Figure 4 jcm-10-01558-f004:**
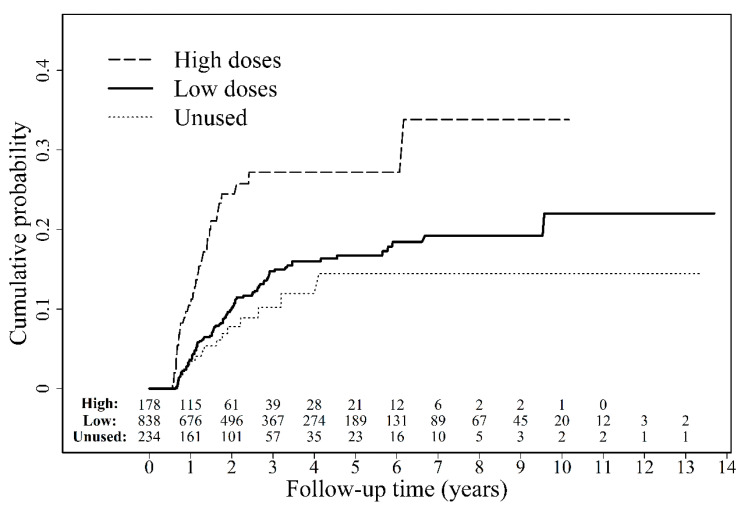
Effects of hepatocellular carcinoma recurrence on mortality of liver transplant recipients. HR, hazard ratio; CI, confidence interval.

**Table 1 jcm-10-01558-t001:** Demographic Data and Clinical Characteristics of the Liver Transplant Recipients in the Study Case (HCC Recurrence) and Control (Non-HCC Recurrence) Groups.

	Recurrence (*n* = 151)		Non-Recurrence (*n* = 1099)		Crude HR	95% CI		*p*
	N or Mean	% or SD	N or Mean	% or SD				
Age ^☨^	53.00	8.24	53.64	9.05	1.000	0.985	1.016	0.9516
Gender								0.9455
Female	117	77.48	853	77.62				
Male	34	22.52	246	22.38	1.013	0.693	1.483	
Cerebrovascular or cardiovascular disease	5	3.31	44	4.00	0.865	0.865	2.134	0.7522
Coronary artery disease	16	10.60	94	8.55	1.451	0.861	2.447	0.1622
Hepatitis B	94	62.25	661	60.15	1.056	0.760	1.467	0.7442
Hepatitis C	40	26.49	337	30.66	0.886	0.618	1.271	0.5113
Liver cirrhosis	140	92.72	1005	91.45	0.912	0.490	1.694	0.7696
Renal failure	3	1.99	38	3.46	0.599	0.186	1.925	0.3897
Pulmonary disease	27	17.88	187	17.02	1.123	0.739	1.706	0.5877
Hypertension	33	21.85	277	25.20	0.969	0.658	1.427	0.8723
Diabetes mellitus	37	24.50	287	26.11	1.011	0.699	1.462	0.9547
Immunosuppressant Treatment								
Cyclosporine	14	9.27	139	12.65	0.626	0.362	1.082	0.0936
Tacrolimus	147	97.35	1054	95.91	1.803	0.666	4.879	0.2458
MMF	135	89.40	881	80.16	1.763	1.051	2.957	0.0316 *
Sirolimus	33	21.85	301	27.39	0.739	0.503	1.084	0.1218

**^☨^** Values are expressed as number of cases and percentage or the mean and standard deviation (SD). Crude hazard ratios (HRs) are provided for liver transplant recipients with HCC recurrence compared with those with non-recurrence after transplantation. HCC, hepatocellular carcinoma; CI, confidence interval. * Represent a statstical significance (*p* < 0.05).

**Table 2 jcm-10-01558-t002:** Annual Number of Patients with De Novo Liver Malignancy in General Taiwanese Populations, and HCC Recurrence in the LTR Cohort from 1998 Through 2012.

Year	General Population		Cohort Population	
	De Novo Liver Malignancy	Population Number	HCC Recurrence	Population Number
1998	7124	21,928,591	0	0
1999	7729	22,092,387	0	4
2000	8101	22,276,672	0	1
2001	8584	22,405,568	0	14
2002	8860	22,520,776	2	15
2003	9404	22,604,550	2	38
2004	9830	22,689,122	8	39
2005	9916	22,770,383	5	52
2006	10,092	22,876,527	6	79
2007	10,110	22,958,360	13	92
2008	10,565	23,037,031	10	142
2009	11,080	23,119,772	19	154
2010	11,023	23,162,123	33	196
2011	11,292	23,224,912	38	252
2012	11,422	23,315,822	15	172
Total			151	1250

HCC, hepatocellular carcinoma.

**Table 3 jcm-10-01558-t003:** Demographic Data and Clinical Characteristics of Liver Transplant Recipients in the Study Case (MMF Use) and Control (MMF Nonuse) Groups.

	MMF Used (*n* = 1016)		MMF Not Used (*n* = 234)		*p*
	*n* or Mean	% or SD	*n* or Mean	% or SD	
Recurrence time (days)	640.40	496.32	595.69	360.24	0.7275
Age ^☨^	53.61	8.42	53.37	11.02	0.7620
Gender					0.3315
Female	222	21.85	58	24.79	
Male	794	78.15	176	75.21	
Hepatitis B	624	61.42	131	55.98	0.1254
Hepatitis C	306	30.12	71	30.34	0.9464
Chronic hepatitis	1009	99.31	231	98.72	0.3585
Liver cirrhosis	949	93.41	196	83.76	<0.0001 *
Hypertension	254	25.00	56	23.93	0.7330
Diabetes mellitus	259	25.49	65	27.78	0.4719
Coronary artery disease	87	8.56	23	9.83	0.5377
Peptic ulcer	601	59.15	126	53.85	0.1379
Renal failure	36	3.54	5	2.14	0.2761
Pulmonary disease	165	16.24	49	20.94	0.0853
Hypercholesterolemia	149	14.67	36	15.38	0.7800
Gout	73	7.19	19	8.12	0.6216
Cardiovascular disease	39	3.84	10	4.27	0.7573

**^☨^** Values are presented as the mean and standard deviation (SD) or number of cases and percentage. * Represent a statstical significance (*p* < 0.05).

**Table 4 jcm-10-01558-t004:** Effects of Hepatocellular Carcinoma Recurrence on Mortality of Liver Transplant Recipients.

	Non-Recurrence (*n* = 1099)	Recurrence (*n* = 151)	Unadjusted			Adjusted		
			Crud HR	95% CI	*p*	Crude HR	95% CI	*p*
High doses	145 (13.19%)	33 (21.85%)	2.265	(1.527, 3.360)	<0.0001	2.234	(1.503, 3.319)	<0.0001
Low doses	736 (66.97%)	102 (67.55%)	-	-	-	-	-	-
Unused	218 (19.84%)	16 (10.60%)	0.742	(0.438, 1.259)	0.2690	0.717	(0.420, 1.226)	0.2240

Adjusted for liver cirrhosis. HR, hazard ratio; CI, confidence interval.

## Data Availability

No new data were created or analyzed in this study. Data sharing is not applicable to this article.

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
