# Peer review of "Effect of Mycophenolate Mofetil Therapy on Recurrence of Hepatocellular Carcinoma after Liver Transplantation: A Population-Based Cohort Study"

_jcm, 2021, doi:10.3390/jcm10081558_

Round 1
Reviewer 1 Report
Dear Authors,
your findings could have a signifcant impact in transplant daily clinical practice. MMF is a current standard therapy in addition to CNI for almost 80% of LTR worldwide. As you know, long term CNI related complications, particularly kidney failure, hypertension and dysmetabolic alterations, are the main cause of LT graft failure. Thus, MMF as CNI minimizing agent represents a critical measure to mitigate these complications. On the other hand, HCC is becaming, more and more, the most prevalent indication to LT and pre-transplant tumor stage (Milan in vs OUT, alfafetoprotein level, biological response to therapy and so on) are regarded as the strongest predictors of recurrence. Moreover, an extension of morphological criteria (up to seven, San Francisco etc) had been progressively applied over time. Your conclusion, who intrinsically lacks of this clinical and biological data as per the type of analysys, would need to be corroborated by these data by showing the cumulative incidence rate of HCC recurrence over years and by including it on multivariate risk factors before this strong information could be licensed.
Author Response
Journal of Clinical Medicine
No: jcm-1132233
The authors wish to thank the reviewer for your helpful comments and suggestions. The manuscript has been substantially improved as the result of your efforts.
Dear Authors,
your findings could have a significant impact in transplant daily clinical practice. MMF is a current standard therapy in addition to CNI for almost 80% of LTR worldwide. As you know, long term CNI related complications, particularly kidney failure, hypertension and dysmetabolic alterations, are the main cause of LT graft failure. Thus, MMF as CNI minimizing agent represents a critical measure to mitigate these complications. On the other hand, HCC is becoming, more and more, the most prevalent indication to LT and pre-transplant tumor stage (Milan in vs OUT, alfafetoprotein level, biological response to therapy and so on) are regarded as the strongest predictors of recurrence. Moreover, an extension of morphological criteria (up to seven, San Francisco etc) had been progressively applied over time. Your conclusion, who intrinsically lacks of this clinical and biological data as per the type of analyses, would need to be corroborated by these data by showing the cumulative incidence rate of HCC recurrence over years and by including it on multivariate risk factors before this strong information could be licensed.
Response: We greatly appreciate these kind comments. Pre-transplant tumor stage and morphological criteria are the well-recognized predictors for estimating post-transplant HCC recurrences. However, this is the intrinsically limitation of our study design to analyze data from a secondary database, National Health Insurance Research database (NHIRD). NHIRD does not contain clinical and biological data. All claim information from the NHIRD enrollment is claimed codes for diagnosis using ICD-9-CM and for treatment prescription. In Taiwan, above mentioned morphological criteria are adopted for patient selection. Only patients fulfill those morphological criteria could receive transplant surgery due to grafts shortage and outcome estimation. Selected suitable recipients will be paid and coded by NHI, and be recruited into our observation cohort. We focused to tailor suitable immunosuppressant combination to reduce their tumorigenesis effects. Here, our data suggests that an MMF-minimizing strategy under a well-balanced combination of immunosuppressants is beneficial for patients with HCC who receiving liver transplantation.
We also mentioned all these limitations on discussion section on Page 12 as following. “Our study presented some limitations. The NHIRD was a secondary database, and many valuable clinical data were unable to be provided, including tumor morphological criteria, histo-pathological information, laboratory studies, physical examination, severity of comorbidities, and the relationship between disease and death. Otherwise, there was no firmly evidence to support the correlation between the dose of MMF and plasma MMF level, and the relationship may be influenced by age, blood albumin level, and comedications. Interpatient variability in plasma MMF level may exist. Further prospective investigation is needed to realize a therapeutic range of MMF levels in blood, and to weigh the immune tolerance for immunosuppression-minimizing strategy. Our study showed that the survival probabilities in group with post-transplant HCC recurrence were lower compared with the group with cured malignancy.”
Reviewer 2 Report
This is an interesting paper because studies on effective immunosuppressive strategies for the management of patients undergoing a liver transplantation (LT) due to hepatocellular carcinoma (HCC) are limited.
In general, mycophenolate mofetil (MMF) possesses anti-proliferative properties effects and is considered in previous reports as to prevent HCC recurrence.
Notably, the results by Tsai et al. demonstrated increased HCC recurrence rate after liver transplantation following MMF administration in a daily dose dependent manner.
However, these are my main concerns:
- What were the HCC tumor sizes in the individual groups (recurrence versus not recurrence group). How many patients were within MILAN or Up to seven criteria?
- Please add detailed histo-pathological information of the tumors (tumor differentiation, microvascular invasion, pre-transplant TACE)
- Was an induction therapy with ATG or basiliximab used in the immunsuppressive regime of the patients? How long did the patients receive steroids?
- What were the target levels of CNIs und mtor- inhibitors?
Author Response
Journal of Clinical Medicine
No: jcm-1132233
The authors wish to thank the reviewer for your helpful comments and suggestions. The manuscript has been substantially improved as the result of your efforts.
This is an interesting paper because studies on effective immunosuppressive strategies for the management of patients undergoing a liver transplantation (LT) due to hepatocellular carcinoma (HCC) are limited.
In general, mycophenolate mofetil (MMF) possesses anti-proliferative properties effects and is considered in previous reports as to prevent HCC recurrence.
Notably, the results by Tsai et al. demonstrated increased HCC recurrence rate after liver transplantation following MMF administration in a daily dose dependent manner.
However, these are my main concerns:
- What were the HCC tumor sizes in the individual groups (recurrence versus not recurrence group)? How many patients were within MILAN or Up to seven criteria?
ANS: Thank you for the nice suggestions. The morphological criteria, MILAN or Up to seven criteria, is important selection criteria for transplant candidate with favorable HCC. However, this is the intrinsically limitation of our study design to analyze data from a secondary database, National Health Insurance Research database (NHIRD). NHIRD does not contain clinical and biological data. All claim information from the NHIRD enrollment is claimed codes for diagnosis using ICD-9-CM and for treatment prescription. In Taiwan, above mentioned morphological criteria are adopted for patient selection. Only patients fulfill those morphological criteria could receive transplant surgery due to grafts shortage and outcome estimation. Selected suitable recipients will be paid and coded by NHI, and be recruited into our observation cohort. We focused to tailor suitable immunosuppressant combination to reduce their tumorigenesis effects. Here, our data suggests that an MMF-minimizing strategy under a well-balanced combination of immunosuppressants is beneficial for patients with HCC who receiving liver transplantation.
We also mentioned all these limitations on discussion section on Page 12 as following. “Our study presented some limitations. The NHIRD was a secondary database, and many valuable clinical data were unable to be provided, including tumor morphological criteria, histo-pathological information, laboratory studies, physical examination, severity of comorbidities, and the relationship between disease and death. Otherwise, there was no firmly evidence to support the correlation between the dose of MMF and plasma MMF level, and the relationship may be influenced by age, blood albumin level, and comedications. Interpatient variability in plasma MMF level may exist. Further prospective investigation is needed to realize a therapeutic range of MMF levels in blood, and to weigh the immune tolerance for immunosuppression-minimizing strategy. Our study showed that the survival probabilities in group with post-transplant HCC recurrence were lower compared with the group with cured malignancy.”
- Please add detailed histo-pathological information of the tumors (tumor differentiation, microvascular invasion, pre-transplant TACE)
ANS: This is the limitation of our study design. We revised and mentioned all these limitations on discussion section on Page 12.
- Was an induction therapy with ATG or basiliximab used in the immunsuppressive regime of the patients? How long did the patients receive steroids?
ANS: (1) We found few patients received ATG administration recorded in NHIRD. The reason could be the drugs paid by patient themselves or by research funds. In our analysis, there were 2 patients (1.3%) received ATG in recurrence group, and 5 patients (0.45%) in non-recurrence group. The difference between 2 groups was not statistically significant (p= 0.2106). Otherwise, we found there was no patients received rituximab in recurrence group, and 2 patients (0.18%) in non-recurrence group. No patient claimed the OKT3 prescription in NHIRD. We did not analyze the usage of basiliximab in the immunosuppressive regimen of the patients. (2) Steroids were commonly used medicine in liver transplantation. In recurrence group, there were 135 patients (89.4%) received solumedrol, 105 patients (69.54%) received prednisolone, and 35 patients (23.18%) received methylprednisolone. In non-recurrence group, there were 951 patients (86.53%) received solumedrol, 599 patients (54.5%) received prednisolone, and 170 patients (15.47%) received methylprednisolone. However, we did not analyze how long did the patients receive steroids.
- What were the target levels of CNIs and mtor-inhibitors?
ANS: We found the patient numbers between study case (HCC recurrence) and control (non-HCC recurrence) groups were not statistically significant on the usages of CNIs and mtor-inhibitors. Thus, we did not further analyze the target levels of CNIs and mtor-inhibitors.
Reviewer 3 Report
The authors present a nationwide retrospective cohort on the de-novo development of HCC in LTRs. As a major risk factor for HCC after LTx, MMF usage was found.
The study is well designed, written and concise methods are applied.
Therefore, I have only minor remarks:
- What was the rational for the time definition of 3 months to exclude pts from analyses and categorize them as relaps of primary HCC?
- In Table 1 it is not clear, if crude HRs or the mutlivariate model is presented, and these data should be clarified throughout the whole manuscript
- If the multivariate model is presented, the crude data and HR for MMF and HCC should be provided in the subheading of the Table 1.
- Why did the authors choose the group with low MMF dosages as reference and not the no use group (Table 4)?
Author Response
Journal of Clinical Medicine
No: jcm-1132233
The authors wish to thank the reviewer for your helpful comments and suggestions. The manuscript has been substantially improved as the result of your efforts.
The authors present a nationwide retrospective cohort on the de-novo development of HCC in LTRs. As a major risk factor for HCC after LTx, MMF usage was found.
The study is well designed, written and concise methods are applied.
Therefore, I have only minor remarks:
- What was the rational for the time definition of 3 months to exclude pts from analyses and categorize them as relapse of primary HCC?
ANS: Thank you for the kind comments and suggestions. We excluded patients claimed HCC code within 3 months after liver transplantation by referencing our previous study and others (PMID: 27626495, PMID: 21633099). Many recipients continually stayed in hospital after transplantation and received post-transplant treatment within 3 months. The ICD codes for HCC were sustainably claimed during this administration. This code for HCC was represented as original coding before surgery rather than as a new code for HCC recurrence. Furthermore, the tumorigenesis effects of immunosuppressants take time for development. If the exclusion time interval is too short, it would cause misinterpretation.
- In Table 1 it is not clear, if crude HRs or the multivariate model is presented, and these data should be clarified throughout the whole manuscript
ANS: Many thanks. Our data were presented as crude HRs. We clarified them throughout the revised manuscript.
- If the multivariate model is presented, the crude data and HR for MMF and HCC should be provided in the subheading of the Table 1.
ANS: Our data were presented as crude HRs, not multivariate model.
- Why did the authors choose the group with low MMF dosages as reference and not the no use group (Table 4)?
ANS: MMF was commonly used immunosuppressants (81.28%) in our cohort, and low-dose MMF group was the majority. We focused to find out the appropriate dosage of MMF for safety concern. So, we choose the group of low MMF dosages as reference to figure out a cut-point value.
Round 2
Reviewer 1 Report
Dear Authors,
thanks for your kind efforts in replying to the questions aroused. Take a moment to further stress the intrinsing limitations of this study